# Overcoming barriers in long-term, continuous monitoring of soil CO₂ flux: A low-cost sensor system

Thi Thuc Nguyen[1,2], Nadav Bekin[1,3], Ariel Altman[2], Martin Maier[4], Nurit Agam[3], Elad Levintal[2,*]

[1]Kreitman School of Advanced Graduate Studies, Ben-Gurion University of the Negev, Be'er Sheva, 840071, Israel

[2]Zuckerberg Institute for Water Research, The Jacob Blaustein Institutes for Desert Research, Ben-Gurion University of the Negev, Sde Boker campus, 84990, Israel

[3]French Associates Institute for Agriculture and Biotechnology of Drylands, The Jacob Blaustein Institutes for Desert Research, Ben-Gurion University of the Negev, Sde Boker campus, 84990, Israel

[4]Department of Crop Sciences, Chair of Soil Physics, University of Göttingen, Grisebachstraße 6, 37077, Göttingen, Germany

*Correspondence to:  Elad Levintal (levintal@bgu.ac.il)

**Abmreviations**

CM, chamber method.

Bref, reference $CO_2$ concentration measured by Vaisala $CO_2$ sensor and LI-COR gas analyzer.

$C_{SCD30}$, $CO_2$ concentration measured by low-cost SCD30 $CO_2$ sensors.

$F_{CM}$, soil $CO_2$ flux measured by chamber method.

$F_{GM}$, soil $CO_2$ flux calculated by gradient method.

$F_s$, soil $CO_2$ flux.

GM, gradient method.

LC-SS, low-cost sensor system.

NDIR, non-dispersive infrared.

SD, secure digital.

SWC, soil water content.

**Abstract**. Soil $CO_2$ flux ($F_s$) is a carbon cycling metric crucial for assessing ecosystem carbon budgets and global warming. However, global $F_s$ datasets often suffer from low temporal-spatial resolution, as well as from spatial bias. $F_s$ observations are severely deficient in tundra and dryland ecosystems due to financial and logistical constraints of current methods for $F_s$ quantification. In this study, we introduce a novel, low-cost sensor system (LC-SS) for long-term, continuous monitoring of soil $CO_2$ concentration and flux. The LC-SS, built from affordable, open-source hardware and software, offers a cost-effective solution (~USD700 and ~50 hours for assembling and troubleshooting), accessible to low-budget users, and opens the scope for research with a large number of sensor system replications. The LC-SS was tested over ~6 months in arid soil conditions, where fluxes are small, and accuracy is critical. $CO_2$ concentration and soil temperature were measured at 10-min intervals at depths of 5 and 10 cm. The LC-SS demonstrated high stability during the tested period. Both diurnal and seasonal soil $CO_2$ concentration variabilities were observed, highlighting the system's capability of continuous, long-term, in-situ monitoring of soil $CO_2$ concentration. In addition, $F_s$ was calculated using the measured $CO_2$ concentration via the gradient method and validated with $F_s$ measured by the flux chamber method using the well-accepted LI-COR gas analyzer system. Gradient method $F_s$ was in good agreement with flux chamber $F_s$ (RMSE = 0.15 μmol m$^{-2}$ s$^{-1}$), highlighting the potential for alternative or concurrent use of the LC-SS with current methods for $F_s$ estimation—particularly in environments characterized by consistently low soil water content, such as drylands. Leveraging the accuracy and cost-effectiveness of the LC-SS (below 10 % of automated gas analyzer system cost), strategic implementation of LC-SSs could be a promising means to effectively increase the number of measurements, spatially and temporally, ultimately aiding in bridging the gap between global $F_s$ uncertainties and current measurement limitations.

## 1. Introduction

Soil is the largest terrestrial carbon pool (Lal, 2005). Soil carbon can be subdivided into two general pools: organic and inorganic, with the global storage of each pool at approximately 1,530 and 940 PgC, respectively (Monger et al., 2015). Both organic and inorganic soil carbon exchange with the atmosphere through soil $CO_2$ flux ($F_s$). $F_S$ is one of the largest carbon fluxes in the Earth system (Bond-Lamberty et al., 2018; Friedlingstein et al., 2022). Compared with human-caused increases in atmospheric $CO_2$, annual $CO_2$ efflux from the soil into the atmosphere is much larger (Oertel et al., 2016). Therefore, $F_s$ is considered a crucial carbon cycling metric, important for the determination of an ecosystem's carbon budget, calibration, validation, development of (agro)ecosystem, soil carbon models, and assessment of the current global warming scenarios (Bond-Lamberty et al., 2024; Klosterhalfen et al., 2017; Xiao et al., 2012).

For decades, there has been a lack of $F_s$ monitoring in different parts of the globe. Various initiatives have been undertaken to integrate dispersed $F_s$ observations worldwide into publicly accessible datasets (Bond-Lamberty et al., 2020; Bond-Lamberty and Thomson, 2010; Jian et al., 2021). However, global $F_s$ datasets often exhibit low temporal-spatial resolution and spatial bias (Stell et al., 2021; Warner et al., 2019). These limitations constrain our understanding of the mechanisms governing soil carbon dynamics and bias regional-to-global $F_s$ estimation. The largest uncertainties in $F_s$ estimates are found in tundra and dryland ecosystems primarily situated at the two poles, across Africa, Central Asia, South America, and Australia (Stell et al., 2021; Warner et al., 2019; Xu and Shang, 2016). These gaps can be primarily attributed to logistical constraints in manual data collection and the high costs of commercial measuring devices (Bouma, 2017; Forbes et al., 2023; Xu and Shang, 2016). Addressing logistical and financial constraints is crucial because critical questions concerning carbon dynamics can only be answered through extensive $F_S$ quantification (Kim et al., 2022).

Field methods commonly used worldwide to quantify $F_s$ are the eddy covariance method (Baldocchi et al., 1988; Massman and Lee, 2002), the flux chamber method (CM) (Davidson et al., 2002; Lundegårdh, 1927), and the gradient method (GM) (De Jong and Schappert, 1972; Hirano et al., 2003; Tang et al., 2003). These methods substantially differ in principles, thus deviating in cost and $F_s$ estimation. The eddy covariance method provides $F_s$ from a relatively large surface area (Gu et al., 2012), whereas the CM and GM yield single-point $F_s$ (Bekin & Agam, 2023; Maier and Schack-Kirchner, 2014). The CM allows $F_s$ to be measured directly from the soil surface, while the GM measures subsurface soil $CO_2$ concentration and estimates $F_s$ using Fick's law (Maier and Schack-Kirchner, 2014).

Despite the increasing popularity of the eddy covariance and CM, the GM remains a useful, widely used method (Chamizo et al., 2022; Hirano et al., 2003; Tang et al., 2003; Vargas et al., 2010). In comparison to the other two methods, the GM offers several advantages. First, it mitigates issues associated with eddy covariance, such as turbulence insufficiency, and with CM, such as the microclimate alterations from chamber deployment (Bekin and Agam, 2023; Maier and Schack-Kirchner, 2014). Moreover, GM offers additional insights into the depth profile of gas production, consumption, and exchange in the soil (Maier and Schack-Kirchner, 2014). The most significant advantage of the GM is its lower purchase and

installation costs (1- 2 orders of magnitude less than the CM or eddy covariance method for continuous $F_s$ monitoring).

The development of small, low-cost, low-power, environmental sensors, microcontrollers, and microcomputers has significantly advanced (Chan et al., 2021; Levintal, Suvočarev, et al., 2021). This advancement has led to the extended adoption of low-cost environmental sensing systems in the scientific community (e.g., Helm et al., 2021). Attempts to monitor soil $CO_2$ concentration using low-cost $CO_2$ sensors have been made (Blackstock et al., 2019; Hassan et al., 2023; Heger et al., 2020; Osterholt et al., 2022). Others monitored $CO_2$ fluxes, such as stem, terrestrial, and aquatic fluxes, by implementing the CM using low-cost $CO_2$ sensors and data loggers (Bastviken et al., 2015; Gagnon et al., 2016; Brändle & Kunert, 2019; Carbone et al., 2019; Forbes et al., 2023; Helm et al., 2021). Implementing the GM using soil $CO_2$ concentrations measured by underground $CO_2$ sensors was also reported (Osterholt et al., 2022). However, these studies primarily focused on comparing the precision and accuracy of the low-cost systems with high-end reference systems, typically conducting short-term in-situ examinations lasting from days to weeks, which limits insights into their stability and practicality for long-term use.

To narrow the gap between the uncertainties in the regional-to-global $F_s$ estimations and the capabilities of current measurement methods, in this study, we introduce an open-source, low-cost sensor system (LC-SS) for continuous, long-term monitoring of soil $CO_2$ concentrations and $F_s$. The LC-SS was field-tested over ~6 months in arid soil conditions to examine its stability and accuracy compared to a commercial automated flux chamber. Detailed, step-by-step, do-it-yourself guides describing the design, assembly, and installation are provided to assist non-engineer end-users with easy replication and customization.

## 2. Materials and methods

### 2.1. Hardware

The LC-SS consists of two units: the control unit and the sensing unit (**Fig. 1a** & **Fig. S1**). The control unit includes a microcontroller (Feather M0 Adalogger, Adafruit, USA) accompanied by Secure Digital (SD) card, a latching relay for power control (Latching mini FeatherWing, Adafruit, USA), a clock for accurate time readings (DS3231 RTC, Adafruit, USA), a screen to display real-time results (0.96" 128x64 OLED Graphic Display, Adafruit, USA), and a multiplexer allowing communication to the sensing unit (Gravity 1-to-8 I2C Multiplexer, DFRobot, China). For power, the microcontroller uses a 3.7 V lithium-ion polymer battery (3.7 V 6000 mAh, Adafruit, USA) charged by solar energy via a solar charger (bq24074, Adafruit, USA), and a 6 W 6 V solar panel (Adafruit, USA). The sensing unit includes seven sensors: six $CO_2$ sensors (SCD30, Sensirion, Switzerland, 0-10,000 ppm, accuracy between 400 to 10,000 ppm: ±30 ppm + 3 % of full range), and an atmospheric microclimate sensor (pressure, relative humidity, and temperature, MS8607, DFRobot, China). The SCD30 $CO_2$ sensor also measures temperature and relative humidity (accuracy: ±0.4 °C and ±3 %, respectively).

The LC-SS used in this study featured two waterproof designs of $CO_2$ sensors (**Fig. 1b**): a 50 ml Falcon tube design and a thin coating design. The 50 ml Falcon tube design is an easy-made and long-lasting option, while the thin coating design is suitable for near-surface deployment, effectively reducing errors

associated with measurement depths. Both designs included a hydrophobic membrane to keep water from penetrating the sensor while allowing gas exchange with the surrounding soil. Providing two designs offers end users the flexibility to adopt the option that best fits their needs and accessibility.

The total time required to build and calibrate the LC-SS is ~50 hours, but could vary depending on the user's familiarity with electronics and sensor integration. The detailed do-it-yourself guide of the LC-SS assembly with time estimation for each major step and sensor waterproof designs can be found on our GitHub page (https://github.com/OpenDigiEnvLab/soil-CO$_2$-sensor-system). The hardware details are summarized in **Table 1**.

**Table 1**. Summary of hardware components with examples for potential suppliers (components can be purchased from other suppliers).

| Component | Quantity | Cost (USD) | Sources | Comments |
|---|---|---|---|---|
| Feather M0 Adalogger | 1 | 19.95 | Adafruit | A low-cost, low-power data logger |
| RTC DS3231 with CR1220 battery | 1 | 17.5 | Adafruit | Provides accurate time for the data logger; CR1220 battery should be purchased separately |
| Gravity 1-to-8 I2C Multiplexer | 1 | 6.9 | DFRobot | Enables the connection of multiple CO$_2$ sensors to one data logger |
| 0.96" 128x64 OLED Graphic Display | 1 | 17.5 | Adafruit | For real-time display of measurement results |
| Latching relay FeatherWing | 1 | 7.95 | Adafruit | For power control: programmed to turn on and turn off the system to optimize power consumption |
| P2886A feather header kit | 1 | 0.95 | Adafruit | To connect Feather M0 Adalogger with Latching relay FeatherWing |
| Lithium Ion Battery Pack-3.7 V 6600 mAh | 1 | 24.5 | Adafruit | To provide power for the control and sensor unit |
| Adafruit Universal USB / DC / Solar Lithium Ion/Polymer charger - bq24074 | 1 | 14.95 | Adafruit | To charge the battery using the solar energy from solar panel |
| Medium 6 V-2 W Solar panel | 1 | 29 | Adafruit | |
| SD/MicroSD memory card (8GB SDHC) | | 9.95 | Adafruit | |
| SCD30 CO$_2$ sensors | 6 | 6 ×61.79 | Digikey | 4 sensors with thin coating and 2 sensors with 50ml falcon tube |
| STEMMA QT MS8607 humidity-temperature-pressure sensor | 1 | 14.95 | Adafruit | To measure atmospheric humidity, temperature, and pressure |
| Weather-proof container | 1 | 10 | Local suppliers | For the control unit |
| 3D-printed frame | 4 | 2 | Printed locally | For thin coating of 4 sensors |
| Epoxy | 500 grams | 5 | Local suppliers | For thin coating of 4 sensors |
| Plasti Dip | 50 ml | 5 | Local suppliers | For thin coating of 4 sensors |

| | | ~50 | Local suppliers | |
|---|---|---|---|---|
| Cables, wires, and general equipment:<br>+ 7 × 4-wire cable 3 m (6 for SCD30 sensors and 1 for MS8607 sensor)<br>+ Wires in colors white, green, red, black<br>+ 8 × 4-pin cables with Female Dupont connectors<br>+ 3 × JST PH 2pin cable-male connector<br>+ 1× 4pin PH2.0 cable-male connector<br>+ 2 lever wire connectors<br>+ On/off switch<br>+ Shrinking sleeves of different sizes<br>+ Superglue | | | | |

The hardware is controlled using open-source Arduino code written in C++ (www.arduino.cc). The complete code for the LC-SS can be downloaded from our GitHub page. At every measurement cycle, all sensors are activated, and measurement readings are logged onto the SD card with a corresponding timestamp and displayed on the user screen. The default measurement interval is 10 minutes and can be easily customized if required.

**2.2. Field installation**

The LC-SS was installed at the Wadi Mashash Experimental farm located in the Northern Negev desert of Israel (31°04'14'' N, 34°51'62'' E; 360 meters above sea level). The local climate is arid, with an average annual precipitation of 116 mm, primarily occurring between October and April. The daily average maximum and minimum temperatures in January (winter) are 15.9 °C and 8.0 °C, and in August

(summer) are 33.3 °C and 20.7 °C, respectively. Soil is characterized as sandy-loam loess soil (72.5 % sand, 15 % silt, and 12.5 % clay). Soil organic carbon content between 0-5 and 5-10 cm is 9.37 and 9.13 mg g$^{-1}$, respectively. CaCO$_3$ content between 0-5 and 5-10 cm is 50 and 47 %, respectively.

The LC-SS was installed from 24/05/2023 to 14/11/2023, providing continuous measurements for 175 successive days, spanning both summer and winter. Three CO$_2$ sensors were installed at each depth (5

and 10 cm) to allow comparison and statistical calibration, as detailed in section 2.3. At each depth, two sensors with the thin coating design (labeled as sensor#1_5cm, sensor#2_5cm and sensor#1_10cm, sensor#2_10cm) and one sensor with the 50 ml Falcon tube design (labeled as sensor#3_5cm and sensor#3_10cm) were deployed (**Fig. 1c**). To enable manual gas sampling for field calibration, a 60-cm Polyurethane tube (outer diameter×inner diameter = 6×4 mm) was inserted at each depth. One end of the

tube was aligned with the CO$_2$ sensors, while the other end extended above the soil surface and was sealed with a valve (**Fig. 1d**). Additional measurements included soil water content (SWC) using time-domain reflectometers (TDR-315, Acclima, Inc., USA) installed at 3 and 10 cm depths. Air temperature, atmospheric pressure, and precipitation data were taken from a meteorological station located at the same field where the LC-SS was installed (https://ims.gov.il; Zomet Hanegev station).

$F_s$ measured using the CM $(F_{CM})$ was measured at 1-hour intervals using a non-dispersive infrared (NDIR) gas analyzer (LI-8100A, LI-COR, USA) connected to four automated non-steady-state chambers

(104C, LI-COR, USA). $F_{CM}$ was determined as the average readings obtained from the four chambers. The $F_{CM}$ measurements were conducted for the periods 24/05-18/06, 17-23/08, and 5/9-17/10/2023.

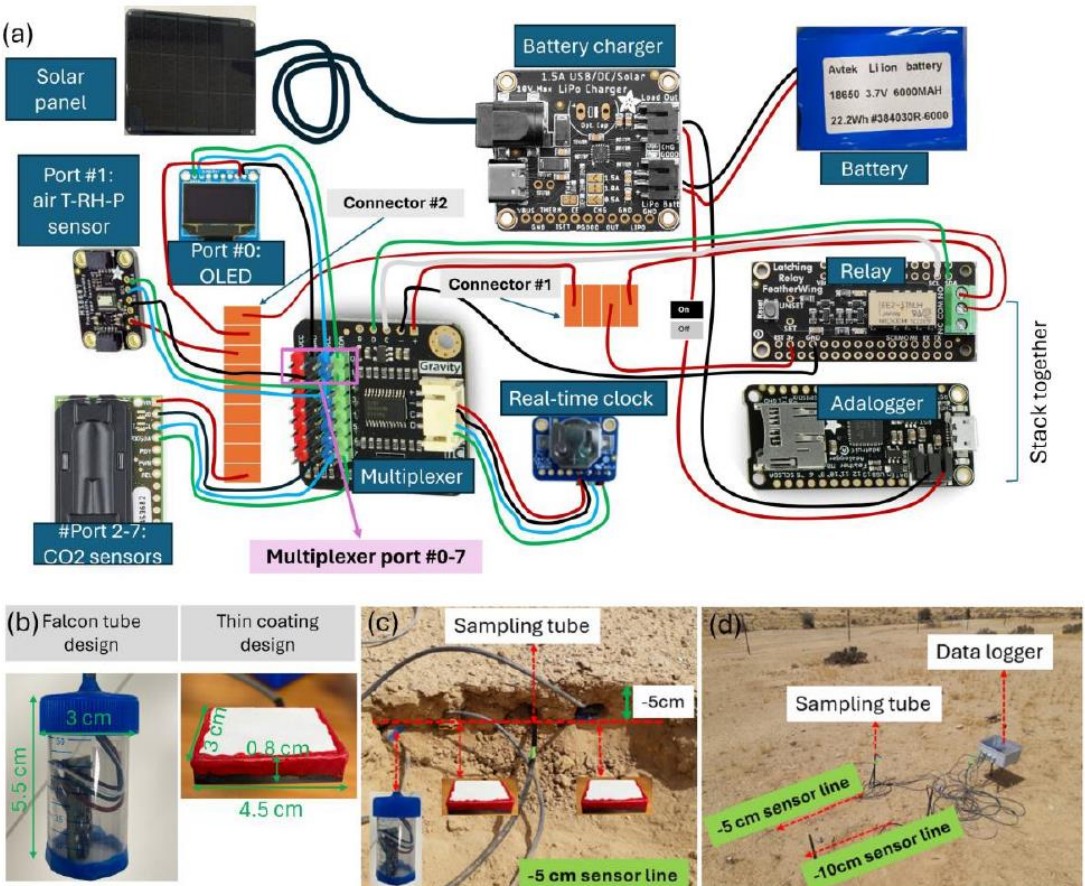

**Figure 1: The design of the low-cost sensor system (LC-SS) (a), two waterproof designs for the SCD30 CO₂ sensors (b), field installation of the CO₂ sensor line at 5 cm (c), and the site after installation (d).**

### 2.3. Two-step calibration of the CO₂ sensors

Calculating $F_s$ based on the GM ($F_{GM}$) (section 2.4) requires accurate soil $CO_2$ concentrations. Therefore, we developed a two-step calibration process for the underground $CO_2$ sensors: a field calibration and a statistical calibration.

For the field calibration, $CO_2$ concentrations from the low-cost SCD30 $CO_2$ sensors ($C_{SCD30}$) were calibrated against reference $CO_2$ concentrations ($C_{ref}$). $C_{ref}$ were obtained by measuring the $CO_2$

concentrations sampled from the sampling tube either by a high-end $CO_2$ sensor (GMP252, Vaisala Inc., Finland) or by LI-COR gas analyzer (LI-8100A, LI-COR, USA) with three replicates from each depth (the choice of calibration devices can be adjusted depending on local availability). $C_{ref}$ by the Vaisala $CO_2$ sensor was measured every 5 hours between 6:00 and 16:00 on two days, 12/06 and 17/07/2023. $C_{ref}$ by the NDIR gas analyzer was measured every 3 hours from 12:00 to 21:00 on 10/9/2023 and from

00:00 to 12:00 on 11/09/2023. In total, the calibration was determined with 21 and 17 measurement

points for each sensor at 5 and 10 cm, respectively, over the range of concentrations from ~300 to ~650 ppm.

Gradual drift was assessed by evaluating whether the pairwise differences in $CO_2$ concentration among three sensors placed at the same depth (sensor#2-sensor#1, sensor#3-sensor#1, sensor#2-sensor#3) changed over time. To quantify this, the pairwise concentration differences were plotted against time, and linear regression was applied to determine the relative drift rate (ppm day$^{-1}$). The cumulative deviation was then estimated as the product of the drift rate and the number of days. If this cumulative deviation exceeded a predefined threshold – set at 10% of the mean concentration in our study – separate field calibration curves were applied to account for the drift.

The statistical calibration consisted of two sequential algorithms. The first algorithm (**Fig. 2a**) addressed abrupt anomalies or jumps of each sensor reading by flagging data points where the difference between measured and smoothed data exceeded 10 % of the measured data point. The smoothed data was executed using the LOESS smoothing algorithm (Jacoby, 2000), which fits multiple locally weighted least squares regressions to estimate a smooth curve through a scatterplot of data points. The second algorithm (**Fig. 2b**) focused on correcting deviation of between three sensors at the same depth, utilizing user-defined thresholds to determine when the difference between one sensor and the other two sensors becomes significant enough to require correction. Thresholds of 5 % and 10 % relative to the average for sensors at 5 and 10 cm, respectively, were defined. All calibration algorithms were applied post-data acquisition, ensuring accurate $CO_2$ concentrations essential for calculating $F_{GM}$.

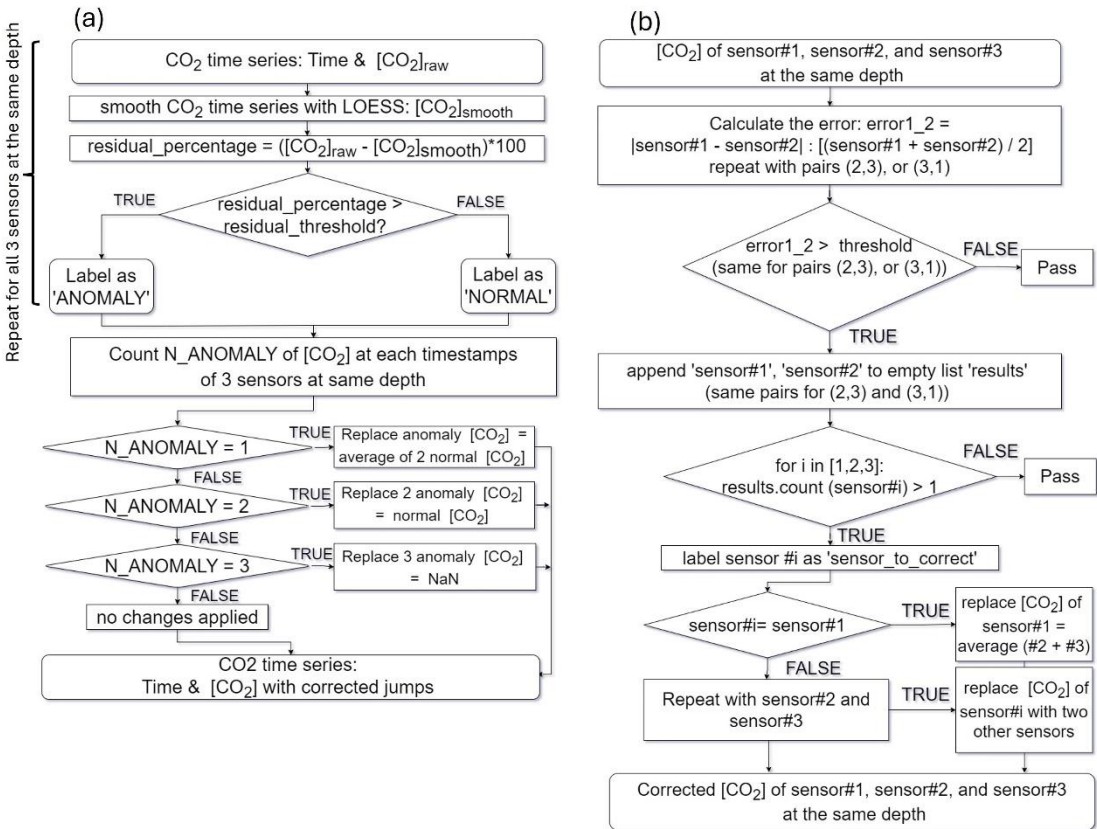

**Figure 2: Flowchart of the two statistical calibration algorithms. The algorithm to correct jumps (a) and the algorithm to correct deviation between three sensors at the same depth (b).**

### 2.4. Calculating the $F_{GM}$ using the LC-SS data

To calculate $F_{GM}$, $CO_2$ concentrations were first corrected for temperature and pressure (**Eq. S1**) and then converted to mole density (**Eq. S2**). The GM is based on Fick's first law, where $F_{GM}$ from depth z to the soil surface is calculated as (De Jong and Schappert, 1972):

$$F_{GM} = -D_s \frac{C_z - C_0}{z} \qquad [1]$$

where $F_{GM}$ [µmol m$^{-2}$ s$^{-1}$] is assumed to be equal to $F_s$ from the soil surface (a positive $F_{GM}$ indicates $CO_2$ efflux and a negative $F_{GM}$ indicates $CO_2$ influx), $D_s$ [m$^2$ s$^{-1}$] is the $CO_2$ diffusion coefficient between depth $z$ [m] (negative) and the soil surface (0 m), $C_z$ [µmol m$^{-3}$] is the $CO_2$ mole density at depth $z$, and $C_0$ [µmol m$^{-3}$] is the atmospheric $CO_2$ mole density ($C_0 = 18741.63$ µmol m$^{-3}$ or 420 ppm). The reference value of 420 ppm was based on the average atmospheric $CO_2$ concentrations measured by a LI-COR gas analyzer between 16/05-18/06 and 2/7-13/8/2023. $F_{GM}$ in this study was calculated using $CO_2$ concentration gradients between 0 and 5 cm depth, as recommended by Chamizo et al. (2022).

The relative $CO_2$ diffusion coefficient in the soil ($D_s/D_a$ where $D_a$ [m$^2$ s$^{-1}$] is the $CO_2$ diffusion coefficient in free air) is estimated based on soil air content-dependent models $M(\varepsilon)$, with $\varepsilon$ being the volumetric air-filled porosity:

$$\frac{D_s}{D_a} = M(\varepsilon) \qquad [2]$$

$D_a$ needs to be corrected to in-situ environmental conditions (Jones, 2013) using **Eq. S3**. Models used in this study to calculate $M(\varepsilon)$, including the most common models, are listed in **Table 2**.

**Table 2: Classical soil diffusion coefficient models used for the GM. Porosity ($\varphi$) values were calculated as described in Eq. S4, and equal to 45%.**

| Authors | Model | Originally developed for |
|---|---|---|
| Buckingham (1904) | $D_s = D_a \varepsilon^2$ | Repacked soils |
| Penman (1940) | $D_s = 0.66 D_a \varepsilon$ | Dry porous materials |
| Millington & Quirk (1961) | $D_s = D_a \dfrac{\varepsilon^{10/3}}{\varphi^2}$ | Different porous materials |
| Millington (1959) | $D_s = D_a \varepsilon^{4/3}$ | Comparison of published results |
| Campell (1985) | $D_s = 0.9 D_a \varepsilon^{2.3}$ | Aggregated silt loam |
| Moldrup (2000) | $D_s = D_a \dfrac{\varepsilon^{2.5}}{\varphi}$ | Unstructured natural soils |
| Marshall (1959) | $D_s = D_a \varepsilon^{1.5}$ | Different porous materials |

| Currie (1970) | $D_s = D_a(\frac{\varepsilon}{\varphi})^4 \varphi^{1.5}$ | Sand |
|---|---|---|
| Lai (1976) | $D_s = D_a\varepsilon^{5/3}$ | Undisturbed and repacked soils |
| Sadeghi (1989) | $D_s = 0.18D_a(\frac{\varepsilon}{\varphi})^{2.98}$ | Soils with clay content from 10.3 to 51.1 % |

From the ten listed diffusion models, ten $F_{GM}$ time series were calculated. The total net flux over the observed period for each $F_{GM}$ time series was calculated by determining the total area under the curve of $CO_2$ efflux minus the total area above the curve of $CO_2$ influx. The average daily cumulative flux [g C m$^{-2}$ day$^{-1}$] was calculated by dividing the total net flux by the total number of days (n = 175).

### 2.5. Validation of $F_{GM}$ using $F_{CM}$

$F_{GM}$ from ten gas diffusion models were validated using measured $F_{CM}$. First, we conducted a cross-correlation analysis (Horvatic et al., 2011) between $F_{CM}$ and $F_{GM}$ to systematically assess the lag time between measured $F_{CM}$ and calculated $F_{GM}$, which reflects the time delay associated with gas transport from the 5 cm depth to the soil surface as previously reported (Sánchez-Cañete et al., 2017). Then, we shifted the $F_{GM}$ using the identified lag time to align with the temporal dynamics of $F_{CM}$.

To evaluate the best-fitted diffusion model, ten shifted $F_{GM}$ calculated based on ten diffusion models were compared with measured $F_{CM}$. The selection of the best-fitted diffusion model is based on a comparison of interquartile range, average daily cumulative flux, r-squared, root mean square errors, and three components of mean squared deviations, namely squared bias, non-unity slope, and lack of correlation (Gauch et al., 2003).

### 3. Results and discussion

Because this study focuses on the development and field performance of the LC-SS for measuring soil $CO_2$ concentrations and calculating $F_{GM}$, our results and discussion will focus mainly on the LC-SS capabilities, such as long-term stability and accuracy.

### 3.1. CO₂ sensors calibration

Over the tested period, we observed a low rate of gradual drift in all six sensors (0.06-0.72 ppm day$^{-1}$) (**Fig. S4**). The cumulative deviations for six sensors were below the predefined threshold - 10% of the mean concentration. Therefore, for the entire period of 175 days, we used one calibration curve for each sensor. The field calibration curves for the six low-cost $CO_2$ sensors are presented in **Fig. 3a**. All sensors show good linearity with high $R^2 > 0.8$. The statistical calibration algorithms (**Fig. 2**) improved both the sudden and permanent drifts (**Fig. 3b**). At 5 cm, only 6.6, 2.1, and 4.4 % out of 25,200 readings of sensors #1 (thin coating), #2 (thin coating), and #3 (falcon tube), respectively, required correction. At 10 cm, 34.5, 1.9, and 1.39 % readings were corrected for sensor #1 (thin coating), #2 (thin coating), and #3 (falcon tube), respectively. Except for sensor#1_10cm, corrections required for other sensors were due to sudden jumps. 34.5% data correction for sensor#1_10 was due to a systematic, permanent drift shifting baseline from ~300 ppm to ~200 ppm from 20/9/2023 until the end of the observed period 14/11/2023.

The results demonstrate the high stability of the CO₂ sensors after 6 months. However, sensor drifting is often system-specific and varies with environmental conditions. Therefore, it is important to detect the gradual drifting of raw data over time (e.g., Fig. **S4**) and conduct field calibration accordingly.

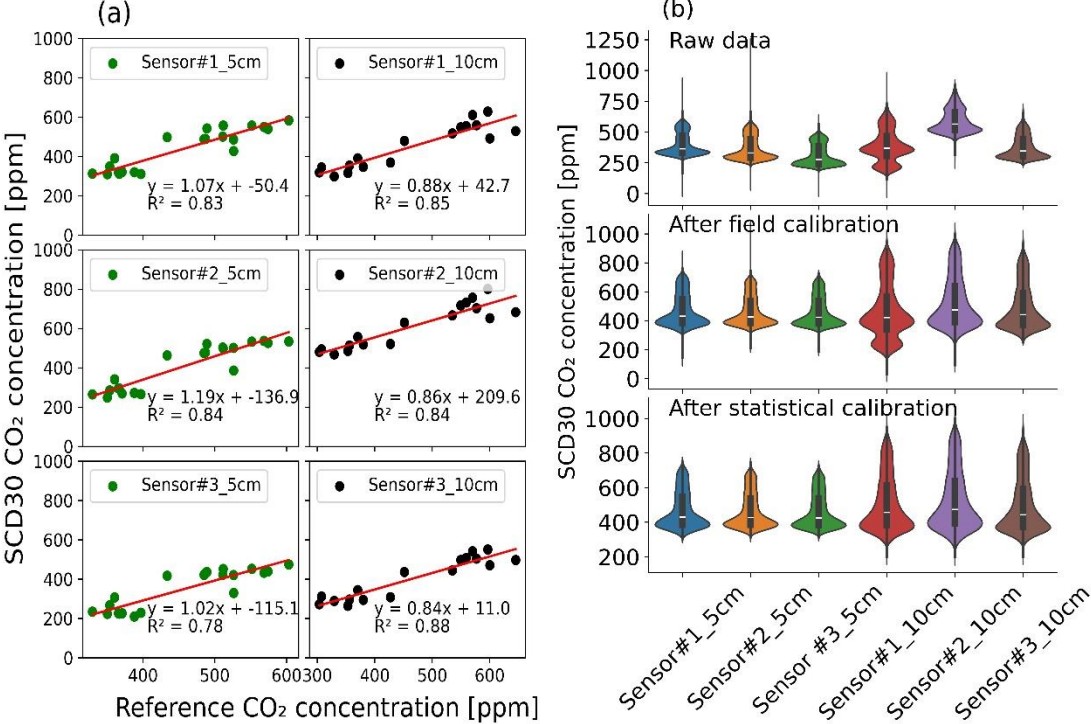

**Figure 3: Calibration curves of the SCD30 CO₂ sensors using reference CO₂ concentration measured by Vaisala CO₂ sensor between 12/6-17/7/2023 and LI-COR gas analyzer 10-11/9/2023 (a), and distribution of CO₂ concentrations collected by six SCD30 CO₂ sensors after field and statistical calibration step (b).**

**3.2. Soil CO₂ concentrations**

The 10-min interval time series of CO₂ concentrations at 5 and 10 cm, and precipitation for one month (24/05-24/06/2023) as an example are shown in **Fig. 4a-b**. The CO₂ concentrations for the entire studied period is presented in the supplementary material (**Fig. S2** and **Fig. S3**). The magnitude of CO₂ concentrations at 10 cm was greater than at 5 cm (~340 – ~730 ppm compared to ~320 – ~1000 ppm, respectively). CO₂ concentrations at both depths during daytime (~7:00 – ~21:00 in summer and ~8:00 – ~19:00 in winter) were higher than in the atmosphere, with average daytime concentrations of 545 and 621 ppm at 5 and 10 cm, respectively. However, during nighttime (all hours excluding daytime hours), soil concentrations were lower than in the atmosphere, with average nighttime concentrations of ~380 ppm at both depths. This indicates an efflux of CO₂ from the soil to the atmosphere during daytime in contrast to an influx of CO₂ from the atmosphere into the soil during nighttime. Daytime efflux and nighttime influx were previously observed in arid soils (Cueva et al., 2019; Hamerlynck et al., 2013; Sagi et al., 2021). The study conducted by Sagi et al. (2021) in the Negev Desert revealed a connection between soil CO₂ influx, cooling soil temperatures, and high soil-to-air temperature gradients,

specifically occurring when SWC was below the threshold of ~8%. We observed similar conditions during our study (**Fig. 4c-e**).

$CO_2$ diurnal cycles at 5 cm showed differences between days with and without precipitation (**Fig. 4c-d**) and between summer months (May-September) and winter months (October-November) (**Fig. 4d-e**). On days with precipitation, the average $CO_2$ concentration increased from 400±20 ppm around 8:00–9:00 to a daily peak of 530±70 ppm at 16:00. On days without precipitation, the morning increase occurred earlier around 11:00–13:00, reaching 662±16 ppm. Inter-season patterns were also observed, with a winter daily peak lower than the summer daily peak by 106±22 ppm. The occurrence of diurnal cycles during all seasons is a typical phenomenon previously reported (Spohn and Holzheu, 2021; Chamizo et al., 2022).

Our results showcase the ability of the underground $CO_2$ sensors to capture typical diurnal and seasonal changes of soil $CO_2$ concentration. The results also highlight the capability of the sensor system to capture "hot moments", such as the effect of precipitation events on $CO_2$ concentration in arid soils, significantly contributing to the understanding of the driving mechanisms underlying these moments.

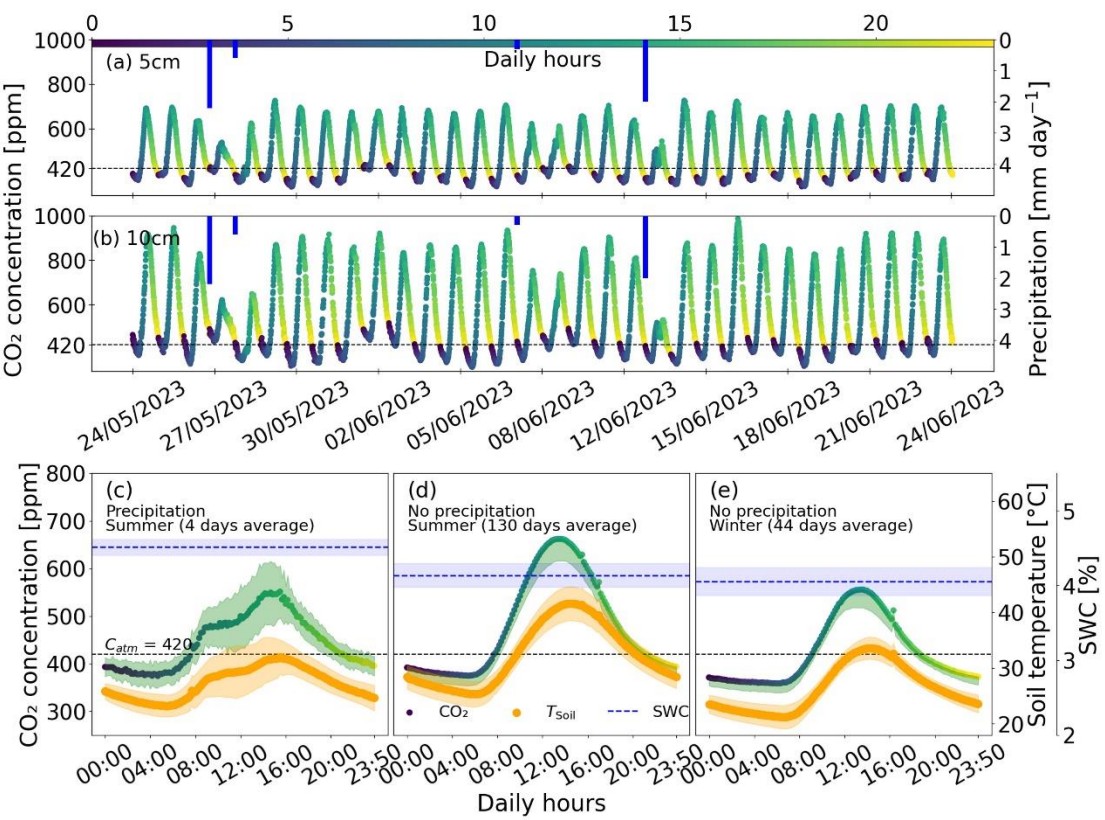

**Figure 4: One month example of continuous $CO_2$ concentration measurements between 24/05-24/06/2023 at 5 cm (a) and 10 cm (b) depths, average daily values at 5 cm of $CO_2$ concentration, temperature, and volumetric soil water content (SWC) during four days with precipitation from May to September (Summer) (c), 130 days without precipitation between May and September (Summer) (d), and 44 days without precipitation between October and November (Winter) (e).**

### 3.3. $F_{GM}$ calculations

The calculated $F_S$ using the GM ($F_{GM}$, **Eq. 1**) and the measured $F_S$ using the CM ($F_{CM}$) are presented in **Fig. S5**; for simplicity, continuous results from only three representative days without precipitation are shown. Calculated $F_{GM}$ using different soil gas diffusion models (**Table 2**) were compared to the $F_{CM}$. We observed a time lag in all calculated $F_{GM}$ compared to the $F_{CM}$. Since the $F_{GM}$ was calculated using the $CO_2$ concentration gradient between 5 cm and the soil surface, $F_{GM}$ can only represent subsurface $F_S$. Cross-correlation analysis was used to evaluate the lag time between the surface $F_{CM}$ and the sub-surface $F_{GM}$ (dashed lines) resulting in a lag time of three hours. To establish temporal alignment between $F_{GM}$ and $F_{CM}$, $F_{GM}$ was shifted three hours to the past (**Fig. S5**, solid lines).

A delay was also observed in the nocturnal influx $F_{GM}$ compared to the nocturnal influx $F_{CM}$. Given the direction of nocturnal $CO_2$ exchange—moving from the atmosphere into the soil—at any given moment, the volume of $CO_2$ traversing a unit surface area at a given time ($CO_2$ influx in units of $\mu mol\ m^{-2}\ s^{-1}$) must exceed that passing through the subsurface region at 5 cm depth. This leads to a more negative nocturnal influx $F_{CM}$ than nocturnal influx $F_{GM}$. Therefore, we used the average daily minimum of nocturnal influx $F_{CM}$ as a reference to shift the magnitude of $F_{GM}$. The time lag between $F_{GM}$ and $F_{CM}$ associated with measurement depth was also reported in previous studies (Sánchez-Cañete et al., 2017); the delay generally increases with sensor depth.

The magnitude and distribution of $F_{CM}$ and $F_{GM}$ (box plots), and average daily cumulative flux (blue scatters) are presented in **Fig. 5a**. The diffusion model evaluation using components of mean squared deviation is presented in **Fig. S6**. In comparison to $F_{CM}$, Buckingham $F_{GM}$ was the most comparable, for both magnitude and distribution, average daily net flux, as well as based on components of mean squared deviation. A representative nine-day time series of Buckingham gradient flux (original and shifted) and chamber flux are presented in **Fig. 5b**. Seven models, including Penman (1940), Marshall (1959), Millington (1959), Millington and Quirk (1961), Currie (1970), Lai (1976), Moldrup (2000) overestimated and two models including, Campell (1985) and Sadeghi (1989), underestimated $F_{CM}$. In generalization, ten models can be classified into two categories based on their assumptions: (1) soil-type/SWC-independent models including Buckingham (1904), Penman (1940), Millington (1959), Campell (1985), Marshall (1959) and Lai (1989) which depends solely on air porosity, and (2) SWC-dependent models including Millington & Quirk (1961), Currie (1970), Sadeghi (1989) which also includes a water-induced linear reduction term, equal to the ratio of air-filled porosity to total porosity ($\varepsilon/\varphi$). The first category can be generalized in the form $b\varepsilon^m$ (with $\varepsilon$ being air-filled porosity, $b$ and $m$ being fitting constants). Currie (1965) has shown that an equation of the form $b\varepsilon^m$ represents well diffusion in dry porous materials, with $m$ typically falling between 1 and 2, and $b$ from 0.5 to 1, depending on the shape of the soil particles. The second category can be generalized in the form $b\varepsilon^m (\varepsilon/\varphi)^n$ (with $b, m, n$ being fitting constants). The addition of the term $\varepsilon/\varphi$, according to Moldrup (2000), helps to better predict diffusion in wet soils. The reasons for the difference of fitting constants ($b, m, n$), for example, Penman (1940) found $b = 0.66$ and $m = 1$, Marshall (1959) $b = 1$ and $m = 1.5$, are that different tortuosity models were used to develop the diffusion model, and the developed diffusion models were validated under varying soils and soil conditions where soil properties such as the pore geometry and the length of gas

passage were different. The majority of models were validated against a wide spectrum of soil texture (e.g., Moldrup 2000 tested on 21 differently textured and undisturbed soils, or Sadeghi tested on 7 soils with clay content 7-51%), fitting constants ($b, m, n$) were therefore concluded as soil-type independent. However, biases were frequently observed, and there is no unique solution holding true for any given specific soil type (Pingintha et al., 2010; Sánchez-Cañete et al., 2017; Yan et al., 2021). For example, in our case, dry, undisturbed soil with 12.5% clay content, matching soil type examined by Sadeghi (1989), Lai (1976), and Moldrup (2000); however, Sadeghi (1989) underestimated $F_{CM}$, while Lai (1976) and Moldrup (2000) overestimated $F_{CM}$. The Buckingham model ($b = 1, m = 2$), one of the models of the first category for dry porous materials, showed the best prediction. However, under higher SWC, increased tortuosity and reduced flow cross-section suggest that higher $m$ in $b\varepsilon^m$ models—or $b\varepsilon^m (\varepsilon/\varphi)^n$ models— may yield better performance. When selecting the most suitable empirical diffusion model for estimating soil gas transport, it is recommended to prioritize $b\varepsilon^m$ models for dry soils and $b\varepsilon^m (\varepsilon/\varphi)^n$ models for wet soils. Testing multiple models in the same category but differing in formulation ($b, m, n$ values) can help assess their sensitivity and applicability to a specific site.

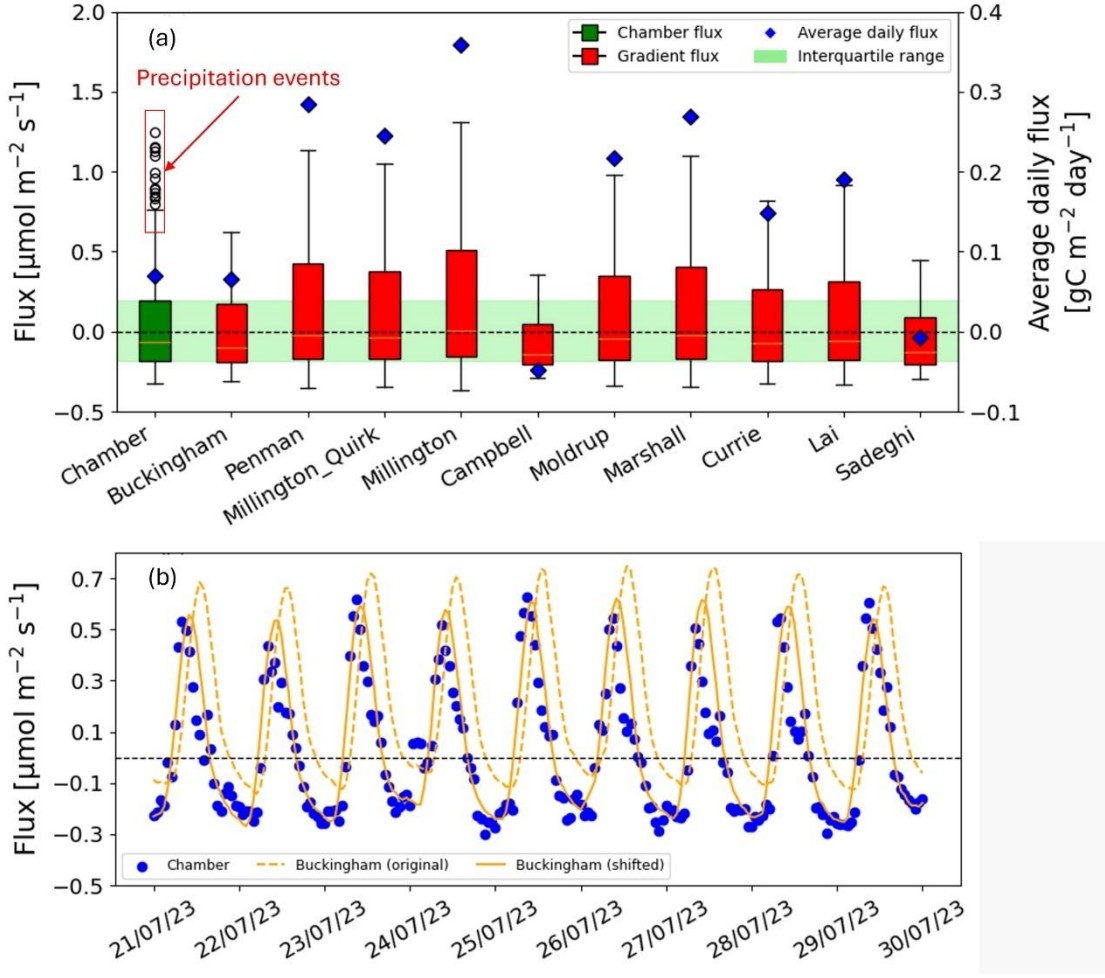

**Figure 5: Comparison of measured chamber flux (green) and calculated gradient flux (red) using ten published gas diffusion models, and average daily cumulative flux (blue scatter) (a), and**

**diurnal cycles of measured chamber flux (blue scatters) and calculated gradient flux using Buckingham diffusion model (dashed orange) and Buckingham gradient flux shifted by 3-hour lag time (solid orange) during nine representative days without precipitation (b).**

The linear regression between Buckingham $F_{GM}$ and measured $F_{CM}$ is presented in **Fig. 6a** ($R^2 = 0.70$, RMSE = 0.15 µmol m$^{-2}$ s$^{-1}$). $F_S$ obtained by these two methods correlated most strongly on days without precipitation (**Fig. 6b**). In contrast, on days with precipitation, large variations between the two methods were observed (outliers in **Fig. 5a & 6a** and **Fig. 6c** – A precipitation event on 13/6/2023 with 2 mm day$^{-1}$). The instantaneous increase of $F_{CM}$ due to precipitation was a well-recognized phenomenon when rewetting occurs in water-limited arid soils (Andrews et al., 2023; Barnard et al., 2020; Fierer & Schimel, 2003). The observed $CO_2$ pulse, as measured by the CM, agrees with the observed pattern of very high rates right after rewetting and slowly declines over time (Kim et al., 2012). These precipitation-induced $CO_2$ pulses were underestimated by the GM. Previous studies also reported that the GM did not capture the abrupt $CO_2$ pulse increases after water application (Jiang et al., 2022; Yang et al., 2018). Rewetting of arid soils after a dry period triggers the sudden increase of microbial activity, leading to a burst in carbon mineralization (Barnard et al., 2020). In arid soil, the top ~1 cm is often the most microbially active due to the presence of biocrust (Weber et al., 2016). The increased $CO_2$ efflux from the topsoil was captured by the CM, yet underestimated by the GM (Jiang et al., 2022; Yang et al., 2018). Under rewetting events, the assumptions of the GM, such as one-directional gas movement and linear concentration gradient with soil depth, are invalid. Greater soil $CO_2$ on the topsoil than in the deeper soil leads to bidirectional concentration gradients and fluxes (Tang et al., 2005). The application of the GM, therefore, is not recommended for $F_S$ estimation of dry soils upon rewetting. It is important to note that this is a well-known methodological limitation, extensively reported in the literature, and it persists regardless of the type of NDIR $CO_2$ sensor used (Fan & Jones, 2014; Tang et al., 2005). Even though $F_{GM}$ under rewetting events is unreliable, it does not limit the application of the GM under relatively steady moisture conditions (i.e., SWC can be moderate to high but no abrupt changes due to rainfall or irrigation) (Fan & Jones, 2014; Turcu et al., 2005).

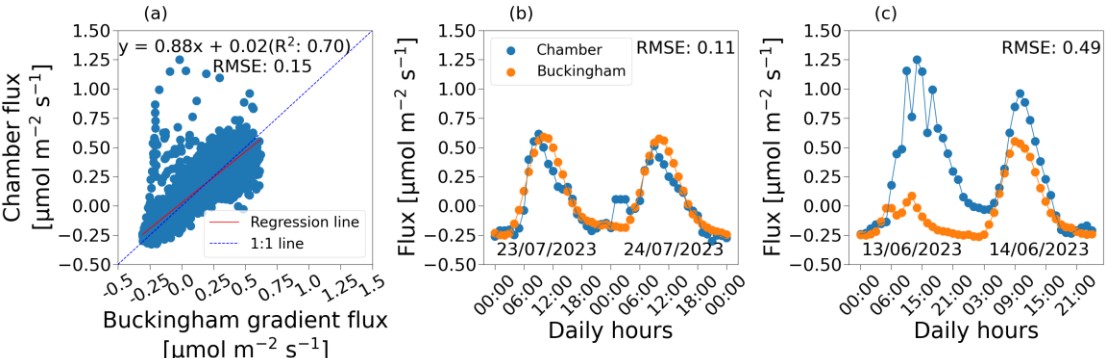

**Figure 6**: **Comparison between the gradient flux ($F_{GM}$) calculated by the best-fitted Buckingham diffusion model and the LI-COR chamber flux ($F_{CM}$) for the whole tested period of 175 days (a), the Buckingham gradient flux (orange) and the LI-COR chamber flux (blue) during two**

**representative days without precipitation (23-24/07/2023) (b), and during two representative days with precipitation (13-14/06/2023) (c).**

### 3.4. Limitations and modifications

The LC-SS system can be built for approximately USD700, taking ~50 hours depending on the user's familiarity with electronics and sensor integration. This relatively low cost and manageable time commitment make the LC-SS a practical and scalable option for long-term, continuous $CO_2$ monitoring, especially in remote or underfunded research settings. However, we acknowledge that this work has certain limitations. The first limitation involves using high-end LI-COR chambers and gas analyzers for the validation of calculated $F_{GM}$. This practice may pose a cost constraint for resource-limited research. Even though using $F_{CM}$ measured by high-end gas analyzers to validate $F_{GM}$ is a recommended practice (Chamizo et al., 2022; Sánchez-Cañete et al., 2017) and applied in this study, it is not inherently obligatory. Several alternatives can be considered. First, the site-specific diffusion coefficient can be measured directly for the calculation of $F_{GM}$ without using published gas diffusion models. For example, Osterholt et al. (2022) suggested an approach to inject $CO_2$ to estimate the diffusion coefficient. Furthermore, high-end, expensive chambers and gas analyzers can also be replaced with a low-cost, open-source chamber system (e.g., Forbes et al., 2023). The same $CO_2$ sensor SCD30, as used in this study, can also be used to manually build a low-cost chamber. When used with the LC-SS, only one chamber-gas analyzer system per several LC-SSs is needed since only a short duration of $F_{CM}$ measurements is required for validation. Additionally, conventional $CO_2$ quantification techniques - such as gas chromatography or the alkali absorption method - can be used to monitor $CO_2$ concentration changes inside a static chamber to quantify $F_S$ (Yan et al., 2021; Pumpamen et al., 2004; Yim et al., 2002; Christiansen et al., 2015). Integrating the LC-SS with the alkali absorption method could be a promising approach that balances affordability, automation, and long-term monitoring of $CO_2$ concentration and $F_S$, while enhancing accuracy; particularly in remote or resource-limited locations where access to high-end instruments like gas analyzers or gas chromatography is not accessible.

The second limitation is that the system was tested only in dry, arid soils. Although a few precipitation events were captured and analyzed, the system's performance under persistently high SWC conditions was not evaluated over the long term. In general, the use of the GM may not be suitable under conditions of sustained soil saturation, frequent rainfall typical of humid climates, or frequent irrigation.

Last, the LC-SS presented here relies exclusively on an SD card for data logging and storage, which requires manual data retrieval and lacks real-time accessibility for monitoring and troubleshooting. Alternatively, we introduce an updated version of LS-SS equipped with a modem for real-time data updates and immediate troubleshooting whenever necessary (e.g., Levintal., et al., 2021). A detailed, step-by-step, do-it-yourself guide for the updated version is also available on our GitHub page.

### 4. Conclusions

This study introduces an innovative LC-SS developed for continuous, long-term monitoring of soil $CO_2$ concentration and $F_s$, facilitating in-situ soil-gas-related research. The LC-SS was built from low-cost, readily available hardware and open-source software components. The LC-SS design emphasizes

modularity, with publicly available, comprehensive, technical documentation for each module, allowing straightforward replication and customization for non-engineering, low-budget end-users worldwide.

The LC-SS was field-tested for ~6 months, showcasing high stability and capabilities to capture the temporal dynamics of soil $CO_2$ concentrations, including diurnal and seasonal variabilities. Furthermore, the agreement observed between the calculated $F_{GM}$ and measured $F_{CM}$, both in the short term (i.e., sub-daily fluctuation) and in the long term (i.e., net $CO_2$ exchange over ~6 months), demonstrate the potential of the LC-SS as a new approach for $F_s$ quantification. The use of LC-SSs and GM is recommended in soils with consistently dry to moderate SWC conditions. For reliable $F_S$ results, the diffusion coefficient can be measured directly, or several methods of $F_S$ quantification (high-end/low-cost chambers, gas chromatography, or alkali absorption method) were suggested for the validation of the calculated gradient flux.

In conclusion, the LC-SS, priced at ~USD700, not only provides high accuracy of $F_s$ but also offers higher temporal resolution and the potential for improved spatial resolution if widely adopted. This, in turn, could contribute to a more comprehensive dataset for regional-to-global estimation of $F_s$ and advancing our understanding of the global soil carbon cycle.

**Data Availability**

The data are available within the above manuscript, the supplementary information, and our GitHub repository.

**Author Contributions**

TTN and EL conceptualized and conducted the study and wrote the first manuscript draft. EL provided the resources. All the authors (TTN, NB, AA, MM, NA, and EL) contributed to the final version.

**Competing interests**

The authors declare that they have no conflict of interest.

**Acknowledgement**

The authors thank Elyasaf Freiman for helping with the field experiments.

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
