# Peer review of "Overcoming barriers in long-term, continuous monitoring of soil CO2 flux: A low-cost sensor system"

_EGUsphere, 2024_

## Author Comment (AC1)

Dear Hirohiko Nagano,

Thank you for your interest in our work.

We would like to answer your questions as follows:

1. *"I have a question regarding the relationship between the diffusion coefficients obtained from the ratio of measured $CO_2$ flux to $CO_2$ conc gradient and from the model. I am unsure, but such evaluation may contribute to considering the limitations and further improving the novel measurement system. What is the relationship between those two diffusion coefficients?"*

We agree that in general, estimating diffusion coefficients obtained from measured $CO_2$ flux and $CO_2$ concentration gradient is a good approach as recommended by, for example, Sánchez-Cañete et al., (2017, 2018).

$D_s$ (in situ $CO_2$ diffusion coefficient [m² s$^{-1}$]) can be calculated as (e.g., Sánchez-Cañete et al., 2018):

$$D_S = -\frac{F_{chamber} * \Delta z}{\Delta C} \quad (1)$$

where:

- $F_{chamber}$ is surface flux measured by chamber method
- $\Delta z = 0.05$ m in our case
- $\Delta C$ is gradient of $CO_2$ molar density

The calculated $D_S$ will be used to optimize a diffusion model specific to the soil of the research site (e.g., Sánchez-Cañete et al., 2018).

$$\frac{D_S}{D_a} = a\theta_a^b \quad (2)$$

where:

- $D_a$ is the diffusion coefficient of $CO_2$ in free air [m² s$^{-1}$]
- $\theta_a$ is the soil air porosity ($\theta_a$ = soil porosity – SWC, where SWC is soil water content)
- a and b are fitting coefficients.

While this approach is theoretically sound, its practical application is sometimes challenging due to limitations in accurately measuring $\theta_a$. To obtain reliable $\theta_a$, reliable SWC measurements are required. However, in our hyper-arid soil (aridity index = 0.07), SWC remains consistently below 5%, which is outside the accuracy range of available moisture sensors. Consequently, $\theta_a$, also suffers from low accuracy. This inaccuracy prevents us from capturing the real-time fluctuations of $\theta_a$, leading to unreliable estimates of $a$ and $b$ when optimizing the diffusion model.

Since improving the accuracy of gradient flux calculations is not the focus of this work, we have not discussed this aspect in detail. However, as requested, Figure 1 below compares $D_s$ values obtained using the chamber-based method ($F_{chamber}$ and $\Delta C$, labeled as $D_{chamber}$) with those derived from the Buckingham model (Buckingham, 1904) (labeled as $D_{Buckingham}$).

[Figure]

**Figure 1.** Comparison between $D_{chamber}$ and $D_{Buckingham}$

2. *"Also, I want to see the relations among measured $CO_2$ flux, $CO_2$ conc gradient, and modeled diffusion coefficient."*

The relations between measured $CO_2$ flux vs. modelled diffusion coefficient and measured $CO_2$ flux vs. $CO_2$ concentration gradient are presented in Fig. 2. From the two correlations, it is shown that flux is mainly driven by $CO_2$ concentration gradient (as expected).

[Figure]

**Figure 2.** Relation between measured $CO_2$ flux vs. Buckingham diffusion coefficient (a) and vs. $CO_2$ concentration gradient (b).

**References**

Buckingham, E. (1904), Contributions to Our Knowledge of the Aeration of Soils, U.S. Dept. of Agriculture, Bureau of Soils, Washington, D. C.

Sánchez-Cañete, E. P., Scott, R. L., van Haren, J., & Barron-Gafford, G. A. (2017). Improving the accuracy of the gradient method for determining soil carbon dioxide efflux. *Journal of Geophysical Research: Biogeosciences*, *122*(1), 50–64. https://doi.org/10.1002/2016JG003530

Sánchez-Cañete, E. P., Barron-Gafford, G. A., & Chorover, J. (2018). A considerable fraction of soil-respired CO2 is not emitted directly to the atmosphere. Scientific reports, *8(1)*, 13518. https://doi.org/10.1038/s41598-018-29803-x

---

## Author Comment (AC2)

Reviewer 1:

Overall, this study presents an interesting and well-written technical paper, showcasing a low-cost alternative to conventional $CO_2$ measuring devices. However, I miss a few critical details, that I think are needed to make it publishable.

First, I think a more detailed evaluation against LI-COR data is needed. Currently, there are few details, and a more in-depth analysis of discrepancies, especially on systematic biases, would be needed. Additionally, it would be good to see if there is temporal drift and to examine in more detail why there are certain times at which there is a relatively large mismatch.

Further, I missed a bit the discussion on how these methods could be applied in low-cost settings. E.g., how accurate are they, if no calibration data from a more expensive system is available? And how cheap are they really, if accounting for the time invested? I think it would be good to give an estimate of the time to assemble it and get it running from scratch. For people wanting to try this, these "time costs" may be very relevant.

Finally, I think it would be important to stress the limitations more clearly. Especially the geographic/climatic scope should be well described. For example, I could imagine that the device would work much worse in a more humid environment, if $CO_2$ evolution after rainfall is a major issue (as you seem to indicate).

We want to thank the reviewer for the positive review and the constructive comments, which helped us improve the manuscript. Detailed answers are given below using blue font. We would like to emphasize the main changes:

(1)     We added a more detailed evaluation to compare the modeled gradient flux with the LI-COR chamber flux. Specifically:
        + An assessment of sensor drift over time.
        + Additional statistics (RMSE, components of mean squared error) for the evaluation of modeled gradient fluxes using measured chamber flux.
        + Additional interpretation for the fitting performance of diffusion models.
        + Discussion on the discrepancies between the best-fitted flux and the measured flux on rewetting events.

(2) We added a discussion on the application of the system in low-cost settings in Section "3.3 Limitations and modifications". Specifically:

+ Suggested alternative low-cost validation methods to the LI-COR chamber flux system.

+ Time cost evaluation is added to the do-it-yourself guide.

+ The geographic/climatic scope for using the device and the gradient method.

First, I think a more detailed evaluation against LI-COR data is needed. Currently, there are few details, and a more in-depth analysis of discrepancies, especially on systematic biases, would be needed. Additionally, it would be good to see if there is temporal drift and to examine in more detail why there are certain times at which there is a relatively large mismatch.

We address the three key points in the above comment:

- First, we analyzed raw data to determine if there was a gradual drift over time and if there was a need for more than one calibration curve for each sensor. For that, we calculated pairwise concentration differences (sensor#2-sensor#1, sensor#3-sensor#1, sensor#2-sensor#3, 3 pairs/depth, 2 depths 5 and 10 cm). The results were added to Figure S4 in the supplementary information (attached below).

  We calculated the relative deviation of each sensor/pair, multiplied by the total number of days, and then compared it to the average concentration. If the relative deviation <= 10% of the average concentration, one calibration curve is needed. Otherwise, separate calibration curves for each period where the relative deviation remains below 10% are required. It is noted that the 10% is the predefined threshold used in this study; depending on the accuracy standard, this threshold can be adopted differently.

  An example of relative deviation calculation:

  For sensors at 5 cm:

  Sensor #2-#1: relative deviation = -0.08*175 = -14 ppm

  Sensor #3-#1: relative deviation = 0.14*175 = 24.5 ppm

  Sensor #2-#3: relative deviation = -0.22*175 = -38.5 ppm

The relative deviations are relatively small compared to the concentration range at 5 cm (300~650) and are also lower than 10% of the average $CO_2$ concentration. Therefore, one calibration curve/sensor was used. We added this analysis to the manuscript with a reference to Figure S4 below.

[Figure]

**Figure S4.** Pair-wise $CO_2$ concentration differences between three sensors at 5 cm (a) and three sensors at 10 cm (b). The drift rate for sensors at 10 cm was evaluated separately for two periods, before and after 20/09/2023, when the baseline of sensor#1_10cm drifted systematically from ~300 to ~200 ppm.

- Second, we added more statistics for the comparison and the selection of the best diffusion model. Specifically, we added components of mean squared deviation: squared bias, non-unity slope, lack of correlation (Gauch et al., 2003), and RMSE. The mean squared deviation analysis is presented in Figure S6 in the supplementary information with a relevant reference in the manuscript.

[Figure]

**Figure S6**. Components of mean squared deviation (MSD) for ten diffusion models. The three components are lack of correlation (LC), non-unity slope (NU), and squared bias (SB). Data for comparison are calculated gradient fluxes ($F_{GM}$) using ten diffusion coefficient models and chamber flux ($F_{CM}$) measured by the LI-COR chamber and gas analyzer.

From the boxplot analysis (new Figure 5a, attached below) and mean squared deviation analysis (Figure S6, attached above), the Buckingham diffusion model showed the best performance. Therefore, it is selected for further analysis to see, on a daily or hourly basis, where the mismatches occurred.

[Figure]

**Figure 5**: Comparison of measured chamber flux (green) and calculated gradient flux (red) using ten published gas diffusion models, and average daily cumulative flux (blue scatter) (a). Diurnal cycles of measured chamber flux (blue scatters) and calculated gradient flux using Buckingham diffusion model (dashed orange) and Buckingham gradient flux shifted by 3-hour lag time (solid orange) during nine representative days without precipitation (b).

- The large mismatches observed when comparing Buckingham modeled gradient flux and measured chamber flux occurred during rewetting events. This is a known methodological limitation of the gradient method and not the focus of this study. We extended the discussion to make this issue clearer.

"The observed $CO_2$ pulse, as measured by the CM, agrees with the observed pattern of very high rates right after rewetting and slowly declines over time (Kim et al., 2012). These precipitation-induced $CO_2$ pulses were underestimated by the GM. Previous studies also

reported that the GM did not capture the abrupt $CO_2$ pulse increases after water application (Jiang et al., 2022; Yang et al., 2018). Rewetting of arid soils after a dry period triggers the sudden increase of microbial activity, leading to a burst in carbon mineralization (Barnard et al., 2020). In arid soil, the top ~1 cm is often the most microbially active due to the presence of biocrust (Weber et al., 2016). The increased $CO_2$ efflux from the topsoil was captured by the CM, yet underestimated by the GM (Jiang et al., 2022; Yang et al., 2018). Under rewetting events, the assumptions of the GM, such as one-directional gas movement and linear concentration gradient with soil depth, are invalid. Greater soil $CO_2$ on the topsoil than in the deeper soil leads to bidirectional concentration gradients and fluxes (Tang et al., 2005). The application of the GM, therefore, is not recommended for $F_S$ estimation of dry soils upon rewetting. It is important to note that this is a well-known methodological limitation, extensively reported in the literature, and it persists regardless of the type of NDIR $CO_2$ sensor used (Fan & Jones, 2014; Tang et al., 2005). Even though $F_{GM}$ under rewetting events is unreliable, it does not limit the application of the GM under relatively steady moisture conditions (i.e., SWC can be moderate to high but not change due to rainfall or irrigation) (Fan & Jones, 2014; Turcu et al., 2005)."

Further, I missed a bit the discussion on how these methods could be applied in low-cost settings. E.g., how accurate are they, if no calibration data from a more expensive system is available? And how cheap are they really, if accounting for the time invested? I think it would be good to give an estimate of the time to assemble it and get it running from scratch. For people wanting to try this, these "time costs" may be very relevant.

- For the application of the low-cost sensor system in low-cost settings, we suggested more methods for validation of the modeled gradient flux.

"Several alternatives can be considered. First, the site-specific diffusion coefficient can be measured directly for the calculation of $F_{GM}$ without using published gas diffusion models. For example, Osterholt et al. (2022) suggested an approach to inject $CO_2$ as a tracer gas to estimate the diffusion coefficient. Furthermore, high-end, expensive chambers and gas analyzers can also be replaced with a low-cost, open-source chamber system (e.g., Forbes et al., 2023). The same $CO_2$ sensor SCD30, as used in this study, can also be used to manually build a low-cost chamber. When used with the LC-SS, only one chamber-gas analyzer system per several LC-SSs is needed since

only a short duration of $F_{CM}$ measurements is required for validation. Additionally, conventional $CO_2$ quantification techniques - such as gas chromatography or the alkali absorption method - can be used to monitor $CO_2$ concentration changes inside a static chamber to quantify $F_S$ (Yan et al., 2021; Pumpamen et al., 2004; Yim et al., 2002; Christiansen et al., 2015). Integrating the LC-SS with the alkali absorption method could be a promising approach that balances affordability, automation, and long-term monitoring of $CO_2$ concentration and $F_S$, while enhancing accuracy; particularly in remote or resource-limited locations where access to high-end instruments like gas analyzers or gas chromatography is not feasible."

- On the accuracy of sensor systems for quantifying $CO_2$ concentration, even for high-end sensors such as the commonly-used Vaisala sensors (https://www.vaisala.com/en/products/instruments-sensors-and-other-measurement-devices/instruments-industrial-measurements/gmp252), field calibration is almost always a must for the insurance of data quality. Therefore, we do not recommend using the raw data without calibration (as mentioned in the manuscript).
- Our low-cost system, ~ 700 USD/per system (with an additional building time of about ~3 days/per system), requires around 1 day every few months for $CO_2$ sensor calibration and flux validation. In our opinion, this can still be considered low-cost. In fact, we have been running five copies of this system along the Negev desert for the last year. This showcases the potential significant opportunities for these types of systems in large-scale, long-term comparative research. In any case, we added the building time to our GitHub DIY guide.

Finally, I think it would be important to stress the limitations more clearly. Especially the geographic/climatic scope should be well described. For example, I could imagine that the device would work much worse in a more humid environment, if $CO_2$ evolution after rainfall is a major issue (as you seem to indicate).

We added to the limitations section the geographic/climatic scope where we think the system will not work well.

"The second limitation is that the system was tested only in dry, arid soils. Although a few precipitation events were captured and analyzed, the system's performance under persistently high SWC conditions was not evaluated over the long term. In general, the use of the GM may not be

suitable under conditions of sustained soil saturation, frequent rainfall typical of humid climates, or frequent irrigation."

Detailed comments:

L46 it would be good to show some evaluation stats for your comparison to LI-COR.

We added the RMSE value (0.15 $\mu$mol m$^{-2}$ s$^{-1}$) to the manuscript.

L59 You might also mention the importance of Fs data for (agro)ecosystem and soil carbon models in calibration, validation, and development.

 We added the above insight and citations to the manuscript.

L145 Since you are talking about $CO_2$ flux, it would be better to give soil organic carbon, not soil organic matter (usually SOM/1.72).

We measured soil organic carbon (9.37 mg/g and 9.13 mg/g for soil at depths 5 and 10cm, respectively). We added the values to the manuscript.

Section 2.5. I think you should do more than just a linear regression with LICOR data to check for the accuracy of your approach. For example, display of relative RMSE, analysis of whether there is a systematic bias, and a slope of the regression different from zero (e.g., Gauch et. Al., 2003, https://www.agronomy.org/publications/aj/abstracts/95/6/1442) are needed for proof of good performance. Also, it may be important to dissect where the systematic differences between sensors in Figure 3b (raw data) stem from.

For the validation of modeled gradient fluxes using measured flux (Section "2.5. Validation of $F_{GM}$ using $F_{CM}$"), besides the comparison using boxplots (Fig. 5a) and the linear regression (Fig. 6a), we added the mean squared deviation components as suggested in Gauch et. al, (2003) (added as Figure S6 in the supplementary) and the RMSE between the measured and the gradient flux modeled by Buckingham equation (added to Fig. 6a) (both figures are attached above). In addition, we added the suggested reference to the manuscript.

The systematic differences between sensors in Fig. 3b raw data occurred most likely because of the automatic baseline correction (ABC) algorithm (as a default algorithm for drift correction in

open-air environmental settings). In soil, since sensors were not exposed to fresh air, so the ABC miscalculated and adjusted each sensor's baseline differently.

L229 I disagree that your displayed results validate the stability after 6 months. They only show how much correction was needed but not if there was a temporal trend in the correction needed (e.g., drift of one sensor from the other two). The latter would be interesting to analyze in detail. In the simplest terms, you could do this by plotting the sensor's raw data over time. Maybe calculate correlation statistics for each month individually.

We agree that more analysis, such as a temporal trend of gradual drift, is necessary to show the system's stability after six months. As mentioned above, we analyzed the relative gradual drift of one sensor compared to the other two at the same depth. The results are shown in Figure S4 in the supplementary section and added to the discussion in section "3.1. $CO_2$ sensor calibration".

"Gradual drift was assessed by evaluating whether the pairwise differences in $CO_2$ concentration among three sensors placed at the same depth (sensor#2-sensor#1, sensor#3-sensor#1, sensor#2-sensor#3) changed over time. To quantify this, the pairwise concentration differences were plotted against time, and linear regression was applied to determine the relative drift rate (ppm day$^{-1}$). The cumulative deviation was then estimated as the product of the drift rate and the number of days. If this cumulative deviation exceeded a predefined threshold - set at 10% of the mean concentration in our study - separate calibration curves were applied to account for the drift."

"Over the tested period, we observed a low rate of gradual drift in all six sensors (0.06-0.72 ppm day$^{-1}$) (Fig. S4). The cumulative deviations for six sensors were below the predefined threshold - 10% of the mean concentration. Therefore, for the entire period of 175 days, we used one calibration curve for each sensor."

Figure 3: More information is needed in the caption so that this figure can stand on its own. E.g., over what time periods were reference and SCD30 measurements made. Additionally, it would be nice to code Fig 3 a by date of the measurements made.

We changed the caption of Figure 3 as suggested to make it stand-alone.

"**Figure 3**: Calibration curves of the SCD30 $CO_2$ sensors using reference $CO_2$ concentration measured by Vaisala $CO_2$ sensor between 12/6-17/7/2023 and LI-COR gas analyzer 10-11/9/2023

(a), and distribution of $CO_2$ concentrations collected by six SCD30 $CO_2$ sensors after field and statistical calibration (b)."

Figure 4: also here, more information is needed so that the figure can stand on its own. E.g., abbreviations are not defined (Fs, Tsoil, SM). Panels c, d, and e should be amended with the exact dates that they refer to. Or are these averages? This is not clear for me.

We changed the caption of Figure 4 as suggested to make it stand-alone. We also added text and error bars to Fig. 4c, d &e to clarify that average data are presented.

[Figure]

**Figure 4**: One month example of continuous $CO_2$ concentration measurements between 24/05-24/06/2023 at 5 cm (a) and 10 cm (b) depths, average daily values at 5 cm of $CO_2$ concentration, temperature, and volumetric soil water content (SWC) during four days with precipitation from May to September (Summer) (c), 130 days without precipitation between May and September (Summer) (d), and 44 days without precipitation between October and November (Winter) (e).

L290 to 291 This part belongs to the Methods section.

This part was moved to section "2.4. Calculating the $F_{GM}$ using the LC-SS data".

Figure 5 a) is quite messy with all the lines and hard to read. I suggest you move this into the supplement and show only the best-fitting method, here. Again, make sure to define all abbreviations so that the figure stands on its own.

We replaced Fig.5a and kept only the best-fitted Buckingham method. A figure with all predicted fluxes is presented in Figure S5 in the supplementary. The caption was changed as suggested to make it stand-alone.

[Figure]

**Figure 5**: Comparison of measured chamber flux (green) and calculated gradient flux (red) using ten published gas diffusion models, and average daily cumulative flux (blue scatter) (a), and diurnal cycles of measured chamber flux (blue scatters) and calculated gradient flux using Buckingham diffusion model (dashed orange) and Buckingham gradient flux shifted by 3-hour lag time (solid orange) during nine representative days without precipitation (b).

L306 do you mean "correlated most strongly"?

 Yes, the sentence was corrected to "correlated most strongly".

Figure 6. Please also report additional evaluation statistics, as suggested above. And again, please define all abbreviations to make the figure stand on its own.

We added RMSE as additional evaluation statistics to Fig. 6a and changed the caption to make it stand-alone.

[Figure]

**Figure 6**: Comparison between the gradient flux ($F_{GM}$) calculated by the best-fitted Buckingham diffusion model and the LI-COR chamber flux ($F_{CM}$) for the whole tested period of 175 days (a), the Buckingham gradient flux (orange) and the LI-COR chamber flux (blue) during two representative days without precipitation (23-24/07/2023) (b), and during two representative days with precipitation (13-14/06/2023) (c).

Section 3.3 is a bit short and I missed a clear recommendation where your system is to be used and where it could have limitations. For example, does it only work well in arid regions (your mismatch mainly occurring on rainy days could suggest this)? Additionally, what methods could be used to improve the reliability of your system? What cheaper methods to double-check your results could you recommend? Could your system even be applied in a manually built chamber to conduct the chamber method?

As mentioned above, we added to the manuscript alternative methods, such as the alkali absorption method, as a cheaper and more accessible method to double-check the modeled gradient flux and also added more discussion on the limitations and a clearer recommendation of where and when the system may/may not work well.

L345 this is the first time you mention maintenance requirements. It should be in the results/discussion if you want to have it in the conclusion.

We removed it from the conclusions section.

L350 I think you should still mention the limitations and potential geographic/climatic suitable ranges (i.e., you only tested it in a very arid environment).

We fully agree – please see our previous answers above.

---

## Author Response (AR1)

We appreciate the time and effort of the reviewers and the editor in providing constructive feedback to improve the manuscript.

To be consistent with the first revision, we would like to address the comments of three reviewers in the order: Reviewer 1, Reviewer 2 (numbered the same as in the first revision), and the new Reviewer 3.

Reviewer 1 asked that you

(i) Provide a more detailed evaluation against LI-COR, especially on systematic bias, sensor drift, and temporal mismatches,

(ii) Discuss use in low-cost settings, including accuracy without calibration and total time/cost,

(iii) Clarify geographic/climatic limitations,

(iv) Improve figure captions, definitions, and visual clarity

(v) Provide additional evaluation metrics and offer clearer guidance on limitations and recommendations.

Your responses seem to address most of the major comments, but there is room for improvement in the quantification of costs at least in the discussion.

We added the quantification of costs, specifically the time cost. Changes were made in the abstract, materials and methods, and limitations and modifications sections:

Abstract section: "The LC-SS, built from affordable, open-source hardware and software, offers a cost-effective and time-manageable solution (~USD700 and ~50 hours for assembling and troubleshooting), accessible to low-budget users, and opens the scope for research with a large number of sensor system replications."

Materials and methods section: "The total time required to build and calibrate the LC-SS is ~50 hours, depending on the user's familiarity with electronics and sensor integration. The detailed do-it-yourself guide of the LC-SS assembly with time estimation for each major step and sensor waterproof designs can be found on our GitHub page (https://github.com/OpenDigiEnvLab/soil-$CO_2$-sensor-system)."

Limitations and modifications section: "The LC-SS system can be built for approximately USD700, taking ~50 hours depending on the user's familiarity with electronics and sensor integration. This relatively low cost and manageable time commitment make the LC-SS a practical and scalable option for long-term, distributed $CO_2$ monitoring, especially in remote or underfunded research settings.

Reviewer 2 asked for additional information on

(i) The diffusion coefficients

(ii)    The relationship between flux, $CO_2$ gradient, and modeled diffusion, which I believe you partially addressed

(iii)    The generalizability of dCO2-dominance, which you have not addressed. Please do.

We carefully read again comments from reviewer 2, particularly the part: "*A more detailed discussion comparing the assumptions and empirical bases of each model in relation to the site's soil texture, structure, and water content would strengthen the interpretation. Specifically, elaborating on why models like Campbell and Sadeghi underestimate fluxes at this site, likely due to their development in structured or clay-rich soils and their strong attenuation of diffusion when air-filled porosity is low. Or why other models, like Millington or Marshall, tend to overestimate flux in coarse, dry soils by overemphasizing the role of air-filled porosity. These insights could assist researchers in choosing suitable diffusion coefficient models for various environments. This is particularly relevant in situations where users do not have access to flux chambers for validation. In such cases, selecting diffusion models based on soil texture and structure could improve results.*". In addition, the editor also specified "*The generalizability of dCO2-dominance, which you have not addressed. Please do.*" Besides adding more discussion on diffusion models as did in the first revision, we added a general guideline for selecting the most suitable empirical diffusion model for estimating soil gas transport:

"When selecting the most suitable empirical diffusion model for estimating soil gas transport—particularly for $CO_2$ flux—the general guideline is to prioritize models developed and validated under similar soil conditions. Additionally, testing multiple models that utilize the same input variables (for example, total porosity and air-filled porosity) but differ in formulation can help assess their sensitivity and applicability to a specific site."

We admit we are not sure that this was the full meaning of item #3 in the comment above; however, this is what we understood, and therefore added the section above to the manuscript.

Reviewer 3 asks that you

(i)    Provide a deeper discussion of why the Buckingham model works well in arid, sandy soils,

(ii)    Compare empirical assumptions of other diffusion models in the context of soil texture and moisture,

(iii)    Evaluate and discuss the performance of other models during wetter periods, not just Buckingham

(iv)    Discuss application in the absence of chamber validation (model selection based on soil properties).

I see you have responded to most of these, but there is a clear lack of a wet-conditioned model comparison across multiple models. This must be thoroughly addressed.

We understand reviewers 2 and 3 shared similar comments as both mentioned "a clear lack of wet-conditioned model comparison across multiple models" and that this is "a missed opportunity."

This comment stems from the mismatch observed during wetter conditions following precipitation events. While sporadic precipitation events occurred during the study period, these did not result in sustained moderate to high soil water content. As such, the conditions were insufficient to support a comprehensive evaluation of the performance of multiple empirical diffusion models under wetter soil regimes. We added an explanation for the observed mismatch between modeled and measured fluxes following precipitation events in our first revision (L381-393) and acknowledged the limitation that our system was tested under dry, arid soil conditions (L433-436). However, to make it clearer, we also emphasized this issue in the abstract:

"Gradient method $F_s$ was in good agreement with flux chamber $F_s$ (RMSE = 0.15 µmol m$^{-2}$ s$^{-1}$), highlighting the potential for alternative or concurrent use of the LC-SS with current methods for $F_s$ estimation—particularly in environments characterized by consistently low soil water content, such as drylands."

---

## Author Response (AR2)

Dear Authors,

Thank you for your revisions. I believe you have addressed most of the previous comments. However, there are a couple of points that remain unresolved, particularly the comment from R2, which you noted was somewhat unclear.

As I interpret R2's intention, they are requesting that you:

1. Expand the discussion of model assumptions and how these align (or not) with the soil texture, structure, and moisture conditions at your site.

2. Explain model-specific biases, such as why certain models may under- or overestimate diffusion. This is likely linked to point 1.

3. Contextualise model performance within your specific site conditions and address whether the observed dominance of $CO_2$ flux is applicable to other environments.

4. Discuss the generalisability of the observed $CO_2$ dominance. Can this finding be expected in other soil types, climates, or land management settings?

You don't need to provide extensive additional text, these points are interrelated and can likely be addressed with some concise but focused discussion. Still, incorporating them will strengthen the discussion and better align your manuscript with R2's expectations.

Raphael VISCARRA ROSSEL

Dear Raphael,

We would like to thank you for your patience in reviewing this work.

To answer your comments, we have thoroughly read all the original articles where each of the models used in this study was first introduced. To the best of our understanding, it is not obvious to explain why one model is overestimated or underestimated the measured flux from our soil (or any specific soil). Diffusion models were developed based on different tortuosity assumptions that are hard to measure or validate. Diffusion models, in general, can belong to two categories: (1) soil-type and SWC-independent model, typically in the form $b\varepsilon^m$ for dry porous media, or (2) SWC-dependent with the addition of the water-induced linear reduction term $(\varepsilon/\varphi)^n$ for wet porous media (with $b, m, n$ being fitting constants and depending on tortuosity). Models of both categories were developed by fitting data from soils of different textures to solve for $b$, $m$, and $n$; therefore, they are highly empirical. If one model of the form $b\varepsilon^m$ is overestimated, we can only say that $m$ is too large or tortuosity is over-emphasized. Thus, we should test a few models of the same category to find the best fit.

We replaced the previous discussion on the diffusion models L335-361 with the text below:

"Seven models, including Penman (1940), Marshall (1959), Millington (1959), Millington and Quirk (1961), Currie (1970), Lai (1976), Moldrup (2000), overestimated and two models, including Campell (1985) and Sadeghi (1989) underestimated $F_{CM}$. In generalization, ten models can be classified into two categories based on their assumptions: (1) soil-type/SWC-independent models including Buckingham (1904), Penman (1940), Millington (1959), Campell (1985), Marshall (1959) and Lai (1989) which depends solely on air porosity, and (2) SWC-dependent models including Millington & Quirk (1961), Currie (1970), Sadeghi (1989) which also includes a water-induced linear reduction term, equal to the ratio of air-filled porosity to total porosity ($\varepsilon/\varphi$). The first category can be generalized in the form $b\varepsilon^m$ (with $\varepsilon$ being air-filled porosity, $b$ and $m$ being fitting constants). Currie (1965) has shown that an equation of the form $b\varepsilon^m$ represents well diffusion in dry porous materials, with $m$ typically falling between 1 and 2, and $b$ from 0.5 to 1, depending on the shape of the soil particles. The second category can be generalized in the form $b\varepsilon^m (\varepsilon/\varphi)^n$ (with $b, m, n$ being fitting constants). The addition of the term $\varepsilon/\varphi$, according to Moldrup (2000), helps to better predict diffusion in wet soils. The reasons for the difference of fitting constants ($b, m, n$), for example, Penman (1940) found $b = 0.66$ and $m = 1$, Marshall (1959) $b = 1$ and $m = 1.5$, are that different tortuosity models were used to develop the diffusion model, and the developed diffusion models were validated under varying soils and soil conditions where soil properties such as the pore geometry and the length of gas passage were different. The majority of models were validated against a wide spectrum of soil texture (e.g., Moldrup 2000 tested on 21 differently textured and undisturbed soils, or Sadeghi tested on 7 soils with clay content 7-51%), fitting constants ($b, m, n$) were therefore concluded as soil-type independent. However, biases were frequently observed, and there is no unique solution holding true for any given specific soil type (Pingintha et al., 2010; Sánchez-Cañete et al., 2017; Yan et al., 2021). For example, in our case, dry, undisturbed soil with 12.5% clay content, matching soil type examined by Sadeghi (1989), Lai (1976), and Moldrup (2000); however, Sadeghi (1989) underestimated $F_{CM}$, while Lai (1976) and Moldrup (2000) overestimated $F_{CM}$. The Buckingham model ($b = 1$, $m = 2$), one of the models of the first category for dry porous materials, showed the best prediction. However, under higher SWC, increased tortuosity and reduced flow cross-section suggest that higher $m$ in $b\varepsilon^m$ models—or $b\varepsilon^m (\varepsilon/\varphi)^n$ models—may yield better performance. When selecting the most suitable empirical diffusion model for estimating soil gas transport, it is recommended to prioritize $b\varepsilon^m$ models for dry soils and $b\varepsilon^m (\varepsilon/\varphi)^n$ models for wet soils. Testing multiple models in the same category but differing in formulation ($b, m, n$ values) can help assess their sensitivity and applicability to a specific site."